# Preclinical safety evaluation of continuous UV-A lighting in an operative setting

Rachael Guenter[1], Rui Zheng-Pywell[1], Brendon Herring[1], Madisen Murphy[1], Jeremy Foote[2], Kevin Benner[3], J. Bart Rose[1]*

1 Department of Surgery, University of Alabama at Birmingham, Birmingham, Alabama, United States of America, 2 Department of Microbiology, University of Alabama at Birmingham, Birmingham, Alabama, United States of America, 3 GE Current, a Daintree Company, East Cleveland, Ohio, United States of America

* jbrose@uabmc.edu

## Abstract

### Background

Germicidal ultraviolet (UV-C) light has been shown as an effective modality for disinfection in laboratory settings and in the operative room. Traditionally, short-wavelength UV-C devices, which have previously been shown to cause DNA damage, are utilized only for disinfection in pre- and post-operative settings and are not continuously active during operations. Continuous use of intraoperative UV light has potential to decrease pathogens and subsequent surgical site infections (SSIs), which arise in approximately 5–15% of operative cases. SSIs are a significant determinant of patient morbidity, readmission rates, and overall cost. Therefore, a method of UV light disinfection with a low risk of DNA damage is needed so that greater antimicrobial protection can be afforded to patients during the entirety of their surgical procedures. A new disinfection device that harnesses longer-wavelength UV-A light to disinfect the surgical field throughout the entirety of the procedure, including pre- and post-operation has been developed.

### Methods

This study aimed to determine if UV-A light administered intraoperatively was safe, as defined by the minimal presence of DNA damage and safe amounts of reflection upon medical personnel. Using *in vitro* models, we examined the differential impacts of UV-C and UV-A light on DNA damage and repair pathways. In a murine model, we looked at the production of DNA damage photoproduction in relation to UV-A versus UV-C exposure.

### Results

Our results show UV-A light does not induce a significant amount of DNA damage at the cellular or tissue level. Furthermore, a preclinical porcine study showed that surgical personnel were exposed to safe levels of UV-A irradiance from an overhead UV-A light used during an operation. The amount of UV-A transmitted through surgical personal protective equipment (PPE) also remained within safe levels.

**Data Availability Statement:** All relevant data are within the manuscript and its Supporting Information files.

**Funding:** J.B.R. received a research grant from GE Current that funded this study. K.J.B. is an employee of GE Current, a Daintree company, and has filed intellectual property on behalf of GE Current, a Daintree company and General Electric Company that pertains to aspects of this work. The funders had assisted with study design, data collection and analysis, decision to publish, or preparation of the manuscript.

**Competing interests:** The authors have declared that no competing interests exist.

## Conclusions

In conclusion, we found that UV-A may be safe for intraoperative use.

## Introduction

Approximately 30 million patients undergo surgical procedures each year in United States, and up to 5% of these patients will experience surgical site infections (SSIs), with infection rates as high as 15% for some operations [1]. SSIs are significant determinants of morbidity, readmission, and cost to the healthcare system [2]. Studies have also shown that patients who develop an SSI face greater financial burden from surgery when compared to uninfected patients [3]. Additionally, patients with an SSI experienced prolonged hospitalization stays, which contributed to negative outcomes on their physical and mental health [3]. The need for surgical procedures continues to increase, highlighting the urgent need for better SSI prevention practices to reduce medical complications and exacerbated costs [3, 4]. Several interventions have been proven to reduce SSIs, including appropriate antibiotic use, normothermia maintenance during procedures, and minimizing intra-abdominal bowel contamination [1, 5]. Despite these interventions, the rate of SSIs continues to rise globally [6].

The use of germicidal ultraviolet (UV) light has been shown to be an effective modality for disinfection of inanimate entities, such as room air and surfaces [7–9]. Ultraviolet-C (UV-C) radiation, encompassing wavelengths ranging from 100 to 280 nm, has been used in healthcare settings to disinfect surfaces in operating rooms, hospital bed rooms, and ambulances [9–12]. The most common UV-C devices use mercury lamps that emit primarily 254nm radiation. UV-C can also successfully sterilize the personal protective equipment (PPE) of healthcare workers [13]. The germicidal properties of UV-C are attributed to its disruption of nucleic acids [13, 14]. Double bonds between the carbon atoms found in pyrimidines and purines are destabilized by UV-C, leading to the formation of dimers in RNA and DNA [15]. Cyclobutyl pyrimidine dimers (CPD) and pyrimidine (6–4) pyrimidone photoproducts (6-4PP) are photoproducts that result from UV-C-induced DNA damage, and these photoproducts interrupt both the cell cycle and DNA replication [9, 16, 17]. This cellular damage ultimately kills bacteria and causes viral inactivation [9, 18–20]. It has also been reported that UV-A can cause DNA damage by inducing oxidative modifications such as 7,8-dihydro-8-oxoguanine (8-oxoG) [21–23]. Oxidixed guanine, 8-oxoG, can be repaired through the DNA base excision repair pathway [24–26]. An enzyme named 8-oxoguanine DNA glycosylase 1 (OGG1) recognizes and excises 8-oxoG modifications [24–26]. OGG1 protects DNA integrity and decreases in OGG1 expression have been associated with oxidative DNA damage [24, 27, 28].

Exposure of UV-C radiation can also cause harmful effects in humans. More specifically, negative effects on the skin and eyes have been reported after UV-C exposure [9]. The ability of UV-C to induce DNA damage is carcinogenic [29]. In fact, UV-C was reported by the World Health Organization to be the most damaging spectrum of sunlight to full thickness skin [9, 30]. Thus, 254nm UV-C at the intensities used by traditional germicidal equipment is only safe for pre- and post-operatively use when humans are not at risk for exposure, although other types of germicidal UV-C emitters exist which may have lower risk [31]. However, skin is differentially affected by the various wavelengths of UV light. UV light can be subdivided into the UV-C spectrum (100–280 nm), UV-B spectrum (280–315 nm), and UV-A spectrum (315–400 nm) [9]. In contrast to UV-C, longer wavelengths of the UV spectrum, particularly UV-A, are able to penetrate deeper layers of the skin (e.g., dermis) [9, 32].

Similar to UV-C, the UV-A spectrum also has antimicrobial properties [33, 34]. UV-A has been shown to indirectly kill cells through the accumulation of free radicals [33, 35, 36]. UV-A

is abundantly present in sunlight and can be germicidal at doses that do not pose a risk to humans [34, 35]. Thus, UV-A could conceivably be a safe option for disinfection in health-care settings. Currently, a UV-A device for use in the operating room is under development and is already shown to successfully reduce pathogens on medical equipment [34]. In North America, the industry standard for human exposure to germicidal lighting is UL 8802, Out-line of Investigation for UV Germicidal Equipment and Systems [37]. The UL standard uses the spectral weighting functions and exposure limits set by IEC International Standard 62471 (Photobiological safety of lamps and lamp systems) and applies them to devices intended for use in occupied spaces [38]. These standards limit human exposure to continu-ous (non-pulsed) UV-A to 10 $W/m^2$. Other North American standards and guidelines such as the Threshold Limit Values published by the ACGIH also set a 10 $W/m^2$ irradiance limit for UV-A [39]. Other standards exist that may set different limits and may be commonly used in Europe or elsewhere [40, 41]. Previous studies using UV-A devices similar to those used in this study found that when irradiance at the head or eye was limited to 10 $W/m^2$, the irradiance at heights roughly equivalent to those of operating tables were approximately 3 $W/m^2$ [34, 42]. Accordingly, we hypothesized that appropriately dosed levels of continuous UV-A irradiance could be used intraoperatively with an acceptable safety profile for both patients and medical personnel. The safety of using UV-A light in the operating room is not known, thus our study was required to begin defining the safety profile of intraoperative UV-A exposure.

## Methods

The UV light devices were provided by GE Current, a Daintree company (East Cleveland, OH). The UV-A emitter was an array of LEDs with peak wavelength of approximately 367nm, and the UV-C emitter was a mercury lamp emitting primarily 254nm radiation. Spectral power distributions for both devices are shown in Fig 1. UV-C and UV-A intensities ($W/m^2$) were measured using a UV light meter (Digi-Sense). All animal experiments were conducted under institutionally approved protocols (IACUC-07733 for *Sus domesticus* and IACUC-21885 for *Mus musculus*).

### Immunoblotting for DNA damange and DNA repair proteins

HEK293 (human embryonic kidney cells) were plated at a density of $10^5$ cells and incubated overnight at 37˚C and 5% $CO^2$ to reach 80% confluence in T-75 flasks. Cells were subsequently exposed to either 1 or 2 hours of either UV-A light at 30 $W/m^2$ (10 times what the UV-A irra-diance may be in a typical application) or UV-C light at 0.3 $W/m^2$. This results in a dose of 10.8 or 21.6 $J/cm^2$ for UV-A and 108 or 216 $mJ/cm^2$ for UV-C. Additional HEK293 cells were plated in the same fashion and were then exposed to either room LED (light-emitting diode) light as a negative control or 10 Gy of gamma radiation over 10 minutes as a positive control. Whole cell lysates were collected 30 minutes after light exposure. Proteins known to be acti-vated via phosphorylation after DNA damage or involved in DNA repair were compared between conditions (pH2A.X, pCHK1, OGG1). Antibody conditions to detect each protein were: pH2A.X (1:500, Cell Signaling Technology #9718), pCHK1 (1:500, Cell Signaling Tech-nology #2348), OGG1 (1:500, Novus Biologicals NB100-1065S), and the loading control GAPDH (1:1000, Santa Cruz sc-47724).

### Alkaline comet assay to detect DNA damage

HEK293 (human embryonic kidney cells) and WI-38 (human lung fibroblasts) were plated to $10^5$ cells and incubated overnight at 37˚C and 5% $CO^2$ to reach 80% confluence in T-75 flasks.

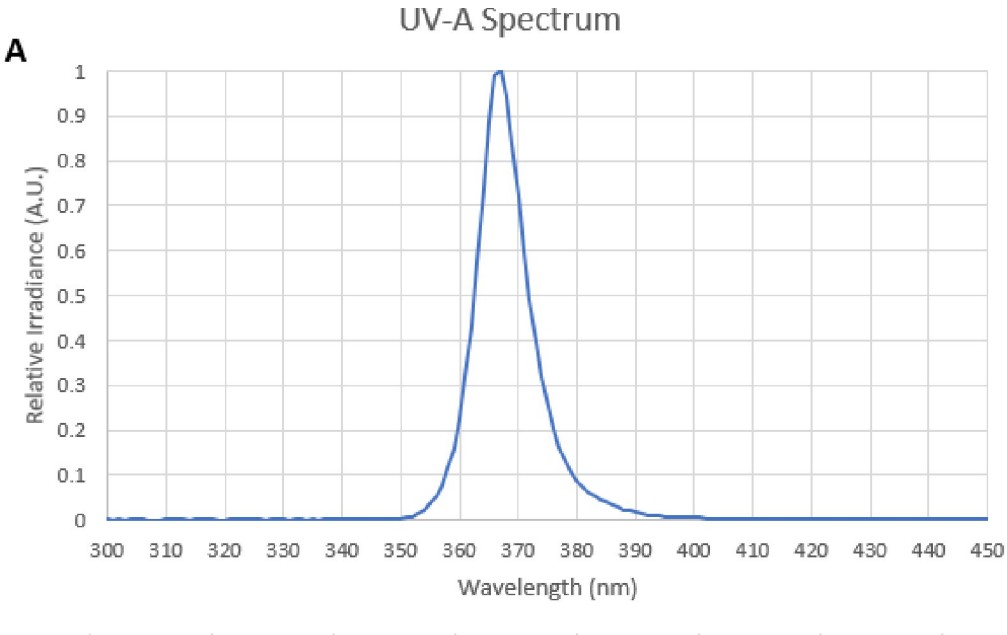

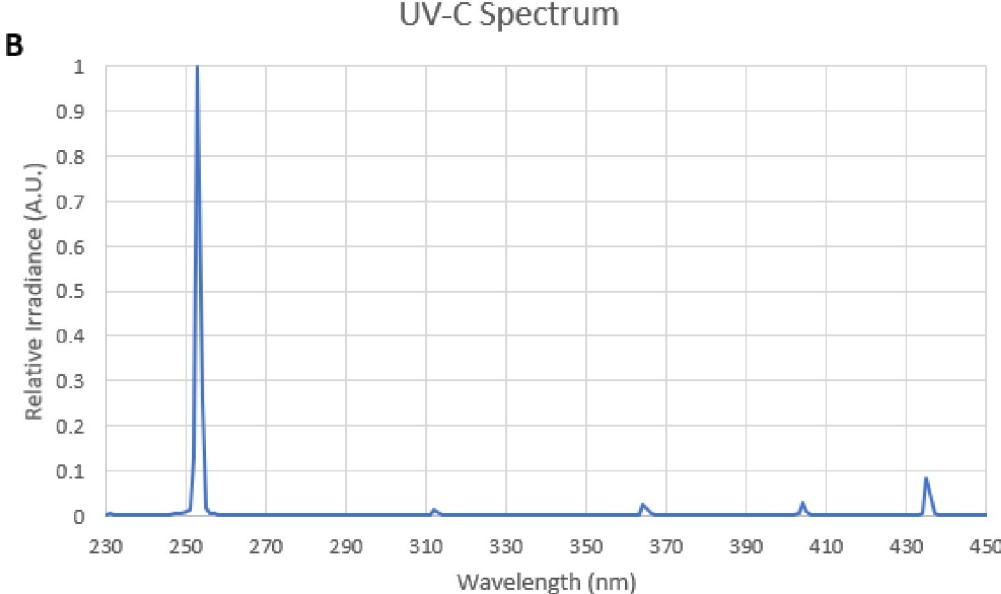

**Fig 1. Spectral power distributions for UV-A and UV-C light sources used in the experiments. (A)** The UV-A source is an array of LEDs with peak wavelength of 367nm and full width at half maximum of 11nm. **(B)** The UV-C source is a mercury lamp emitting primarily monochromatic 254nm radiation.

They were subsequently exposed for 2 hours to either room light, UV-A light at 30 W/m$^2$, or UV-C light at 0.3 W/m$^2$. A CometAssay kit (Trevigen 4250-050-K) was used to process cells. SYBR Green (ThermoFisher S7563) was used for nuclear staining. Comet Assay IV (Instem) software was used to measure tail length and momentum and to calculate % tail DNA. A total of 300 cells were measured for each condition. The Kruskal-Wallis test was used to analyze comet tail length and % tail DNA using IBM SPSS Statistics for Windows, version 25 (IBM Corp, Armonk, NY). Statistical significance was defined as $P < 0.05$.

## Immunofluorescence for DNA damage-related photoproducts

8-week-old C57/BL6 mice were humanely euthanized using a two-step process: (1) unconsciousness was induced via anesthesia (1–2% isofluorane via the nose with inhalation), followed by cervical dislocation. After euthanasia, a liver resection was performed. Resected livers were exposed to either room light (LED), UV-A light at 30 W/m$^2$, or UV-C light at 0.3 W/m$^2$ for a total of 2 hours. Livers were fixed in 4% paraformaldehyde, followed by a sucrose gradient incubation and OCT embedding, prior to cryosectioning. Cryosections were permeabilized with 0.3% triton X-100 and blocked with 5% goat serum for 1 hour at room temperature. After blocking, slides were incubated with the mouse-on-mouse polymer IHC Kit (Abcam ab269452) for 1 hour at room temperature. Liver sections were then incubated overnight in a humidity chamber with primary antibody at 4˚C. Primary antibodies evaluated include: cyclobutene pyrimidine dimers (1:1500, CPD, CosmoBio CAC-NM-DND-001) and pyrimidine-pyrimidone (6–4) photoproducts (1:300, 6–4 PP, CosmoBio CAC-NM-DND-002). The next day, slides were washed with PBS prior to incubating with goat anti-mouse-A488 secondary antibody (1:500, Invitrogen A11001) for 2 hours at room temperature. The slides were then washed in PBS three times before adding DAPI (1:1000) as a nuclear stain. Finally, coverslips were mounted before imaging with the 40X objective on a DM5500B microscope (Leica Microsystems).

## Porcine model for projection of UV-A reflection

A porcine model was used to measure the reflected irradiance of UV-A light. The animal was anesthetized in a supine with a combination of telazol (4.4 mg/kg) and xylazine (4.4 mg/kg), delivered intramuscularly, and then maintained on isoflurane gas. An overhanging UV-A light fixture was set to provide irradiance of 30 W/m$^2$ at the level of the skin (Fig 2A and 2B). Irradiance measurements were obtained at 3 positions 1 foot away from the animal at 3 varying heights (Fig 2C and 2D). UV-A irradiances were measured in triplicates at the designated locations and final values were averaged for two conditions: closed skin and open abdomen.

## Personal Protective Equipment (PPE) model for UV-A protection

To evaluate the level of UV-A protection provided to surgical staff by typical equipment, small samples approximately 50-75mm square were cut from samples of surgical Personal Protective Equipment (PPE) used in the operating environment. Between 4 and 6 samples were taken for each type of PPE. In instances where the PPE was composed of multiple materials, samples were taken of each material type and evaluated separately. A UV/visible spectrophotometer (Lambda 950, PerkinElmer, Inc.) was used to measure spectral transmission of these samples over the full UV-A wavelength range (315-400nm). To evaluate the transmission from the UV-A disinfection device used in this study, an average of spectral transmission over the range of 360-370nm was taken.

## Results

### UV-A light induces no more DNA damage than standard room lighting at the cellular level

To investigate potential effects of UV-A versus UV-C light on DNA damage and DNA reapair mechanisms, we looked at changes in intracellular markers corresponding to damage and repair pathways. Specifically, we assessed protein-level expression of phospho-pH2A.X and phospho-CHK1 (pCHK1) [43, 44] (Fig 3). Cells exposed to 10 minutes of 10 Gy radiation served as a positive control. The phosphorylation of CHK1 (pCHK1) was nearly undetectable

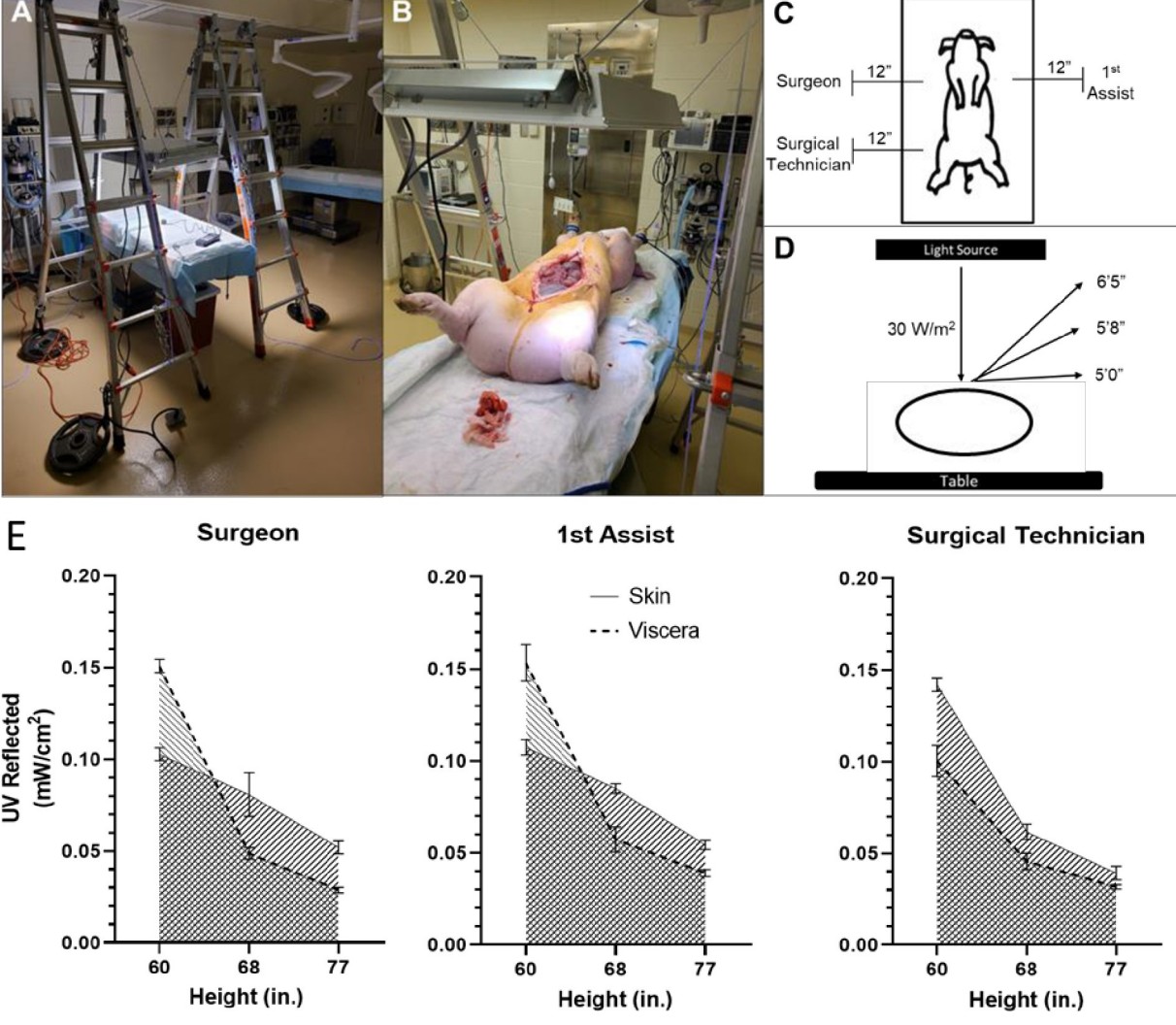

**Fig 2. Model of provider exposure to UV light.** **(A)** Overhanging light to achieve UV-A irradiance of 30 W/m². **(B)** Light source overlying pig with adjustable light source using pulley/lever system. **(C)** Operative team positions around the porcine subject for reflected UV-A measurements. **(D)** Height variables for light measurements. **(E)** Comparison of reflected UV-A irradiance measured at the surgeon, 1ˢᵗ assist, surgical technician locations. Solid line marks reflected irradiance based off skin only measurements. Dotted line represents reflected irradiance based off visceral intensities.

in HEK293 cells exposed to UV-A light for either 1 or 2 hours (Fig 3A). The phosphorylation of histone H2A.X (pH2A.X) is detectable as a 15 kDa band on western blot, which corresponds to the lower band in the pH2A.X blot Fig 3A. Compared to the 10 Gy positive control group, cells exposed to either UV-C light or room light (LED) for 2 hours had phosphorylation of pH2A.X as seen by positive signal from the lower 15 kDa band (Fig 3A). Furthermore, a reduction in the expression of OGG1 has been reported to be correlated with an increase in oxidative DNA damage [27, 28]. Since UV-A exposure has previously been shown to induce oxidative DNA damage, we investigated the expression of OGG1 in HEK-293 cells exposed to either normal LED light (control), UV-A, or UV-C light for either 1 or 2 hours (Fig 3). After 1 hour, cells exposed to UV-A or UV-C had slight increases in OGG1 expression compared to normal LED light (control). Exposure to normal LED light (control), UV-A, and UV-C for 2 hours reduced OGG1 expression in each group compared to the 1-hour exposure groups.

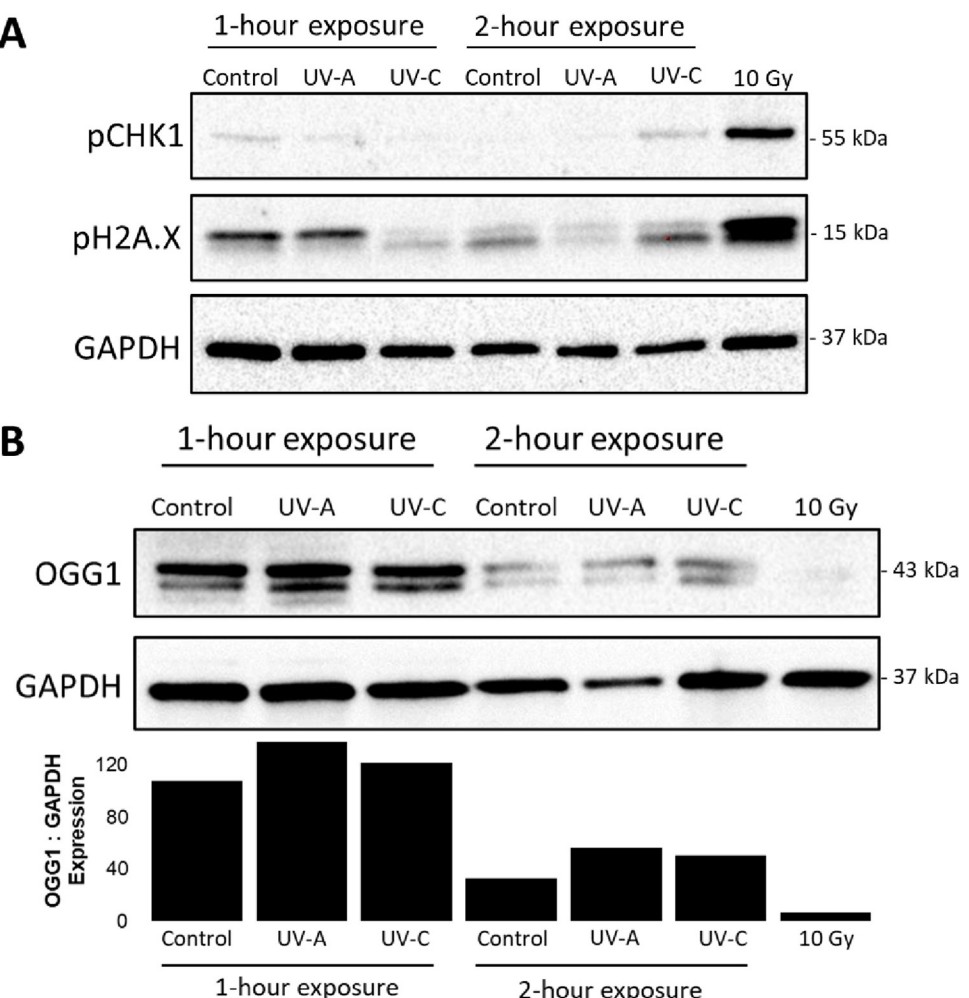

**Fig 3. Western blot comparing markers of DNA repair.** **(A)** HEK293 cells exposed to 10 minutes of 10 Gy radiation as a positive control or UV-C light for 2 hours had detectable amounts of CHK1 phosphorylation (pCHK1). Phosphorylation of histone H2A.X (pH2A.X), the lower 15 kDa band, was highest in the 10 Gy positive control sample. pH2A.X was higher in cells exposed to UV-C for 2 hours compared to cells exposed to either UV-A or normal LED light (control). **(B)** A decrease in OGG1 expression is associated with oxidative DNA damage. Cells exposed to either UV-A or UV-C light for 1 hour had slight increases in expression of OGG1. Exposure to normal LED light (control), UV-A, and UV-C for 2 hours reduced OGG1 expression compared to 1-hour exposure. Cells exposed to 10 minutes of 10 Gy gamma radiation (positive control) had the most significant decrease in OGG1 expression.

Interestingly, cells exposed to normal LED light as a control had the greatest reduction in OGG1 expression, followed by UV-C, and then UV-A had the least reduction in OGG1. As a positive control, cells exposed to 10 minutes of 10 Gy gamma radiation had the most significant decrease in OGG1 expression. This data suggests that at the cellular level UV-A exposure does not significantly induce DNA damage repair pathways.

Further investigation into the impact of UV-A versus UV-C light on DNA damage was done by measuring the amount of double stranded DNA breaks using a Comet Assay. Double stranded DNA breaks were quantified by (1) comet tail length and (2) %tail DNA. In two different cell lines (HEK293 and WI-38), UV-C exposure caused significantly greater tail lengths and %tail DNA compared to a dose of UV-A that was 10-fold higher (Fig 4). This result indicates a higher level of DNA damage, in terms of double stranded DNA breaks, in cells exposed to UV-C versus UV-A light. In both cell lines, the normal room light condition demonstrated

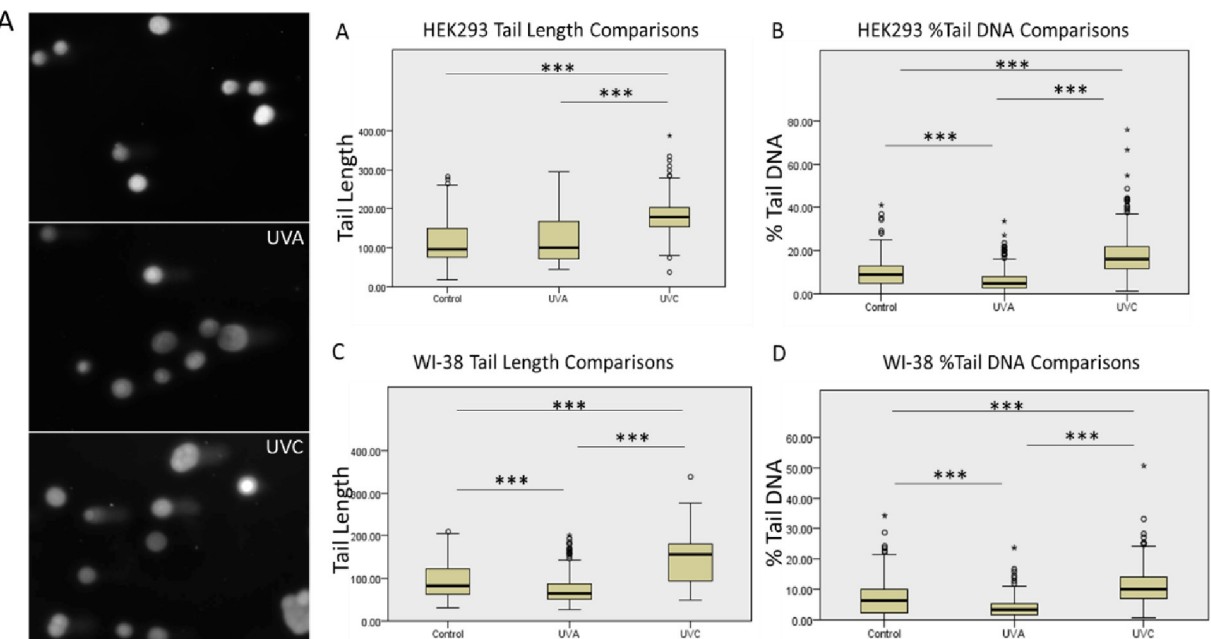

**Fig 4. DNA damage after UV-A/UV-C light exposure assessed by a comet assay.** (A) Representative images of comet assay DNA damage in HEK293 cells exposed to either room light, UV-A, or UV-C light. Comparisons in tail length in (B) HEK293 and (D) WI-38 cells. Comparisons in %tail DNA in (C) HEK293 and (E) WI-38. N = 300, *** indicates a $p < 0.001$.

greater %tail DNA than UV-A exposure, suggesting UV-A light does not induce any more double-stranded DNA breaks than normal room light condition alone.

## UV-A exposure does not produce DNA damage-related photoproducts in tissue

DNA damage caused by UV light exposure creates cyclobutyl pyrimidine dimers (CPD) and pyrimidine (6–4) pyrimidone photoproducts (6-4PP) [9, 16, 17]. These photoproducts accumulate in the nucleus and can lead to cell death. We sought to determine if liver exposed to UV-A light produced CPD and 6-4PP, in comparison to either intraoperative UV-C or the normal room light (LED) condition. We used a mouse model to administer each light group to resected murine livers. Immunofluorescence was performed to detect the presence of CPD and 6-4PP in nuclei after light exposure to the liver (Fig 5). Fluorescence imaging showed UV-A light did not produce either 6-4PP (Fig 5A) or CPD (Fig 5B) in the nuclei of mouse livers, as determined by the absence of fluorescent signal. In contrast, mouse livers exposed to UV-C light did produce both DNA damage photoproducts 6-4PP (Fig 5A) and CPD (Fig 5B), confirmed by positive fluorescent signal. As a negative control, livers exposed to room light (LED) showed no positive signal for 6-4PP or CPD.

## Intraoperative UV-A produces a safe level of reflection on surgical personnel

To determine the reflective irradiance exposure of surgical personnel by UV-A, we employed a porcine model of abdominal surgery (Fig 2). We showed that the greatest amount of reflected UV-A irradiance was 20-fold less than the patient's exposure (0.15 mW/cm$^2$ versus 3.0 mW/cm$^2$). The pattern of reflected UV-A light irradiance did not differ between the surgeon and first assist positions, where the surgeon stood 12 inches to the left of the pig and the first assist

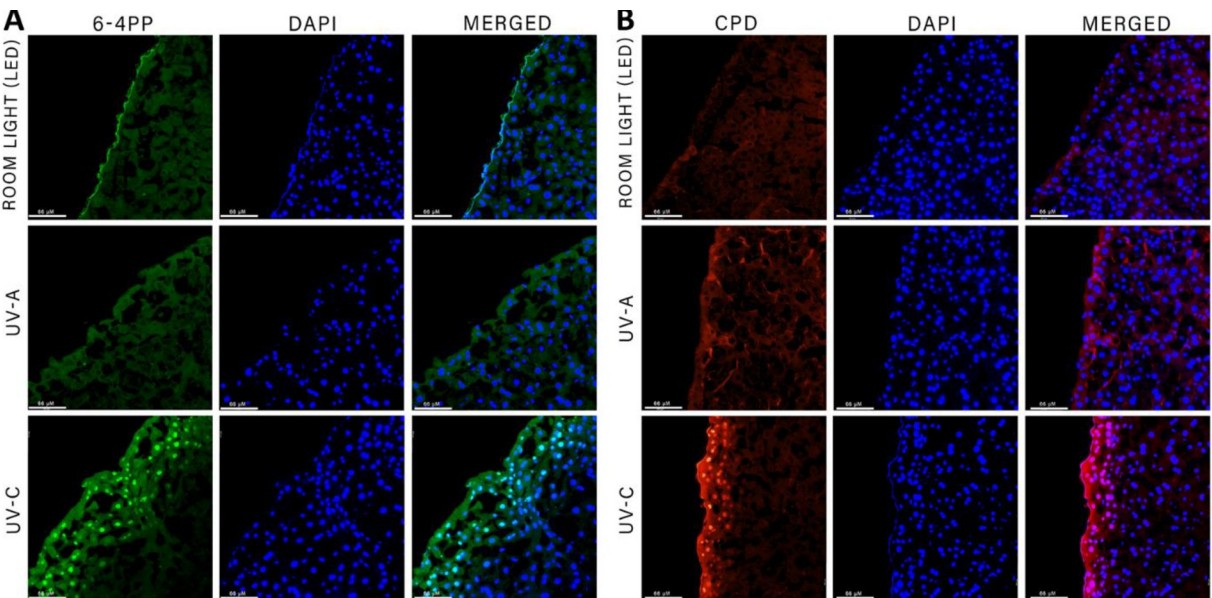

**Fig 5. Comparison of DNA damage photoproducts CPD and 6-4PP in murine liver after either room light, UV-A, or UV-C light exposure.** **(A)** Mouse livers exposed to UV-C light showed positive 6-4PP (green) nuclear staining in the periphery cells, whereas mouse livers exposed to either room light (LED) or UV-A had negative nuclear 6-4PP (green) staining. (B) Positive CPD (red) nuclear signal was only detected in mouse liver exposed to UV-C light. Mouse livers exposed to either room light (LED) or UV-A had negative CPD (red) nuclear staining. DAPI (blue) was used a nuclear stain. Scale bar = 66 μM.

stood 12 inches to the right (Fig 2C). We then measured the amount of UV-A reflection at different personnel heights (60, 68, 77 inches tall) for the surgeon, first assist, and a surgical technican position (Fig 2D). We found that the visceral reflection of UV-A light had a higher irradiance at shorter personnel heights compared to skin exposure, but the visceral reflected irradiance remained lower at taller personnel heights (Fig 2E).

## Analysis of UV-A transmission, transmittance, and absorbance on surgical PPE

To evaluate an additional safety consideration of using UV-A during surgery, we measured the transmission of UV-A light through surgical PPE. Transmittance (transmitted ratio) and absorbance ($-\log_{10}$ of transmittance) were calculated from the measured transmission. We evaluated twelve different types of common surgical PPE, including eyeglasses, surgeon's caps, surgical masks, and surgical gloves. Brand information and reference IDs for each type of PPE are provided in S1 Table. Percent transmission, transmittance ($\tau$, where $\tau$ = %T/100), and absorbance (A, where A = $-\log_{10}\tau$) for the range of 360–370 nm are given below in **Table 1** for the measured PPE. All surgical PPE measured in this study absorbed UV-A at varying levels. Surgical gloves provided the highest level of UV-A attenuation among the measured items, transmitting less than 10% of UV-A in the 360-370nm range. All opaque PPE transmitted 24% or less of these wavelengths, while transparent or mesh fabric PPE transmitted up to 90%.

## Discussion

Surgical site infections (SSIs) pose an ongoing threat to patients' physical and mental health, and the costs of complications associated with SSIs continue to burden the healthcare system. UV light has historically been used as a disinfection agent for inanimate objects. Theoretically,

Table 1. Transmission, transmittance, and absorbance of surgical PPE in the range of 360-370nm.

| Measured PPE | Percent Transmission (%T) of UV-A (360-370nm, %) | Transmittance (τ) of UV-A (360-370nm) | Absorbance (A) of UV-A (360-370nm) |
|---|---|---|---|
| Drape leggings (clear plastic part) | 90 | 0.90 | 0.04 |
| Drape leggings (fabric part) | 10 | 0.10 | 1.00 |
| Eyeglasses | 80 | 0.80 | 0.10 |
| Surgeons Cap (white part) | 73 | 0.73 | 0.13 |
| Surgeons Cap (Blue part) | 19 | 0.19 | 0.72 |
| Bouffant Cap | 64 | 0.64 | 0.19 |
| Light Blue Gown | 24 | 0.24 | 0.62 |
| Dark Blue Gown | 15 | 0.15 | 0.83 |
| Surgical Mask (blue) | 18 | 0.18 | 0.74 |
| Surgical Mask (green) | 17 | 0.17 | 0.78 |
| Drape | 11 | 0.11 | 0.94 |
| Polyisoprene Surgical Glove | 9 | 0.09 | 1.02 |

UV light could be used as a technique to reduce SSIs by maintaining a sterile surgical field, although the safety of intraoperative UV-A light exposure must first be evaluated. UV-C light is used in healthcare settings to disinfect surfaces in the operating room, hospital bed rooms, and ambulances [9–12]. However, UV-C over-exposure is associated with health risks [9, 29, 30]. A safer alternative to typical UV-C light disinfection products could be devices that use low levels of UV-A light. Previous studies have shown that UV-A light is germicidal and is capable of reducing pathogens on steel surfaces [34]. UV-A exposure from an overhead light source significantly reduced the amount of pathogenic microorganisms on medical equipment [34]. The UV-A doses used in this study would result in approximately a 1-$\log_{10}$ reduction of some viruses [34, 45], and up to about a 6-$\log_{10}$ reduction of bacteria on surfaces [33, 34], and the UV-C doses used as a positive control would likewise result in a several-log reduction of many common pathogens [46].

This study aimed to evaluate the safety of using intraoperative UV-A light. Previously, the safety profile of using UV-A light during an operation was unknown, as all commonly used photobiological standards are based on studies of exposed skin or eyes and not internal organs. We found exposure to UV-A light did not cause significant DNA damage. We tested for DNA damage in cells exposed to UV-A light by using a comet assay to detect DNA double stranded breaks, along with western blot to detect expression of DNA damage-related proteins [43, 44, 47]. In further investigation of potential UV-A-induced DNA damage, we did not find any photoproducts (CPD or 6-4PP) formed in the nuclei of internal organ tissue that was exposed to UV-A. These photoproducts are specifically produced by UV exposure, making them a reliable readout for UV-induced DNA damage [48]. Our results investigating DNA damage markers upon exposure to UV-A light are consistent with with a recent study that similarly explored the effects of UV-A light in human cells and murine tissue [49]. In this study, Rezaie et. al found that UV-A exposure on cells did not effect viabilty or induce the DNA damage marker 8-Oxo-2'-deoxyguanosine levels, while significantly reducing a range of pathogens [49]. Taken together, our data and other recent literature suggest that UV-A light does not induce a significant amount DNA damage in cells or tissues.

Our results also showed UV-A reflection in the operating room remained at low levels, thus not posing a health risk to surgical personnel. We found that the visceral reflection of UV-A light had a higher irradiance at shorter personnel heights compared to skin exposure, but the

visceral reflected irradiance remained lower at taller personnel heights. At the position of the surgical technician, there was greater reflective irradiance with the skin exposure at all heights compared to visceral exposure. These differences may be due to differences in UV-A reflectance, specularity, or overall shape between skin and viscera. Furthermore, the amount of reflected UV-A surgical personnel experience when UV-A was adminstered overheard during a procedure was less than 20-fold of what was experienced by the patient. The industry photobiological standards limit continuous UV-A exposure to 10 W/m$^2$. If a system is designed to limit exposure at the patient to this level, the surgical staff will experience over an order of magnitude less irradiance than the patient. Furthermore, it is also important to consider reflected UV-A onto surgical staff because the eye is more sensitive to UV-A than the skin. We would expect, if using an overhead UV-A disinfection light, that the highest irradiance would be at the top of the surgeon's head and at a much lower irradiance at the eye (looking downward at the table or horizontally). Overall, the amount of UV-A irradiance reflected onto surgical personnel was below industry photobiological standards limits of 10 W/m$^2$ for continuous UV-A exposure, which indicated that overhead UV-A exposure during an operation did not pose a health risk.

In our study of UV-A absorption by surgical PPE, each item measured absorbed some UV-A. In practice, a ceiling-mounted UV-A disinfection system that abides by accepted industry photobiological safety standards would not expose humans to levels of UV-A that may be hazardous [38, 39]. While UV-A-absorbing PPE is not necessary, using such may increase confidence in the safety of a UV-A by surgical staff who would be frequently exposed to germicidal wavelengths.

Although we examined multiple aspects of UV-related DNA damage and intraoperative UV-A reflection, the studies described herein had several limitations. First, we did not investigate all molecular pathways of DNA damage. Implementation of overhead UV-A devices in procedures would naturally require that the other equipment be adequately integrated. In future studies, it will be imperative to investigate the potential ability of UV-A light to reduce SSIs, as confirmation and successful implementation of this modality could greatly benefit patient care. Overall, our findings suggest that UV-A light can safely be used in the operating room.

## Supporting information

**S1 Fig. Raw western blot images from Fig 3. (A)** Raw blot images from Fig 3A with ladders shown, the rightmost lane was the negative control, untreated HEK293 cells and was not used in the final figure for publication. **(B)** Raw blot images from Fig 3B with ladders shown, the rightmost lane was the negative control, controls of HEK293 exposed to UV in a biosafety hood for 2 hours and untreated HEK293 cells were not used in the final figure for publication. (PDF)

**S1 Table. Brand information and reference IDs for PPE used in the UV-A transmission, transmittance, and absorbance study.** (DOCX)

## Author Contributions

**Conceptualization:** Kevin Benner, J. Bart Rose.

**Data curation:** Rachael Guenter, Rui Zheng-Pywell, Brendon Herring, Madisen Murphy, Jeremy Foote, Kevin Benner, J. Bart Rose.

**Formal analysis:** Rachael Guenter, Rui Zheng-Pywell, Brendon Herring, Jeremy Foote, Kevin Benner, J. Bart Rose.

**Funding acquisition:** Kevin Benner, J. Bart Rose.

**Investigation:** Kevin Benner, J. Bart Rose.

**Methodology:** Kevin Benner, J. Bart Rose.

**Project administration:** Kevin Benner, J. Bart Rose.

**Resources:** Kevin Benner, J. Bart Rose.

**Supervision:** Kevin Benner, J. Bart Rose.

**Validation:** Kevin Benner, J. Bart Rose.

**Visualization:** J. Bart Rose.

**Writing – original draft:** Rachael Guenter, Brendon Herring, Kevin Benner, J. Bart Rose.

**Writing – review & editing:** Rachael Guenter, Rui Zheng-Pywell, Brendon Herring, Madisen Murphy, Jeremy Foote, Kevin Benner, J. Bart Rose.

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
