## [Decision Letter · Decision Letter 0]

27 Jan 2023

PONE-D-22-32617Preclinical safety evaluation of continuous UV-A lighting in an operative settingPLOS ONE

Dear Dr. Rose,

Thank you for submitting your manuscript to PLOS ONE. After careful consideration, we feel that it has merit but does not fully meet PLOS ONE’s publication criteria as it currently stands. Therefore, we invite you to submit a revised version of the manuscript that addresses the points raised during the review process.

We look forward to receiving your revised manuscript.

Kind regards,

Cihun-Siyong Alex Gong, Ph.D.

Academic Editor

PLOS ONE

Journal Requirements:

"Conflicts of interest to disclose include that J. Ba Rose (JBR) received a research grant from GE Current that funded this study. Co-author Kevin Benner (KJB) is an employee of GE Current, a Daintree company, and has filed intellectual property on behalf of GE Current, a Daintree company and General Electric Company that pertains to aspects of this work."

6. We note that Figure 5 in your submission contain copyrighted images. All PLOS content is published under the Creative Commons Attribution License (CC BY 4.0), which means that the manuscript, images, and Supporting Information files will be freely available online, and any third party is permitted to access, download, copy, distribute, and use these materials in any way, even commercially, with proper attribution. For more information, see our copyright guidelines: http://journals.plos.org/plosone/s/licenses-and-copyright.

a. You may seek permission from the original copyright holder of Figure 5 to publish the content specifically under the CC BY 4.0 license. 

Additional Editor Comments:

It looks like your research is technically sound. However, both the referees have raised several concerns regarding your studies. Please revise your manuscript thoroughly before resubmitting it.

Reviewers' comments:

Reviewer's Responses to Questions

**Comments to the Author**

1. Is the manuscript technically sound, and do the data support the conclusions?

Reviewer #1: Partly

Reviewer #2: Yes

2. Has the statistical analysis been performed appropriately and rigorously? 

Reviewer #1: N/A

Reviewer #2: Yes

3. Have the authors made all data underlying the findings in their manuscript fully available?

Reviewer #1: Yes

Reviewer #2: Yes

4. Is the manuscript presented in an intelligible fashion and written in standard English?

Reviewer #1: Yes

Reviewer #2: Yes

5. Review Comments to the Author

Reviewer #1: Authors investigated the safety of UVA-lighting system in an operative room in vitro and in vivo study of porcine model, concluding that UVA irradiation in an OR might be safe for patients and medical personnel. It is certainly promising results, but some major points need to be clarified.

Major points;

1. Authors should show the wavelength specification of UVA lamps they used in this study. The information is including wavelength distribution, peak wavelength, with filtration system or not? etc. If it contains UVB ranged wavelength that have surely harmful potential effects as genotoxicity. Furthermore, most important point is that UVA lamps actually has germicidal efficacy? Authors should show the efficacy data of lamps both UVA and UVC lamps they utilized in this study.

2. DNA damage evaluation; UVA could induce a number of ROS related modification, as authors described in the text, including DNA damage as 8-oxoguanie (8-oxoG) that could potentially induce genotoxicity. It means that the evaluation of 8-oxoG staining after UVA exposure is more important than comet assay which commonly assayed direct DNA damage by UVB or UVC.

Minor points;

3. Figure 4. “Negative stain”; Is this isotype antibody as only second antibody OR no exposure of either UV lamps?

4. Figure 4. 6-4PP/Room light; too high background, should be modify the immunohistochemical conditions.

5. Authors should have mentioned about the latest trend of disinfection devices such as Far-UVC lamps which have a property (222nm-emission) of high germicidal efficacy and no risk of skin carcinogenesis and now be replacing by the conventional UVC disinfection system.

Reviewer #2: The manuscript present an study on the effects of UVA and UVC radiation on DNA damage using in vitro and murine models. Analysis of the irradiance provided by an overhead UVA light developed for being used during a surgical procedure is also presented. The analysis includes data obtained at different height and data of the light reflected by a porcine model to mimic the reflection s produced during an standard surgical operation.

During the revision of the state of the art, I found a work (Rezaie A, et al., Ultraviolet A light effectively reduces bacteria and viruses including coronavirus. PLoS One. 2020 Jul 16;15(7):e0236199. doi: 10.1371/journal.pone.0236199. Erratum in: PLoS One. 2020 Aug 11;15(8):e0237782. PMID: 32673355; PMCID: PMC7365468.) were the authors show an study of the effects of UVA radiation on mammalian (mice) internal visceral cells. They exposure colonic mucosa to UVA and examined through endoscopy the mucosa in order to detect possible damages. Besides the authors performed tissue analysis.

We think that this paper should by included in the reference section and discussion by comparing the results obtained in the present work with those included in the suggested reference.

I think that the paper needs to include data on the dose. Irradiance data is important designing the UVA light and I thank the authors for providing this data. But from a disinfection point of view, the dose is the most important data. Most of the papers related with UVA, UVB or UVC radiation provides the dose (J/m2 or mJ/cm2). So, I suggest the authors to include data on the dose used in their experiments. For calculating the dose it would by necessary to indicate the exposure time. Besides it would be interesting to know the exposure time of the tissue during a common surgical operation.

In the same direction, I think that it is important to study the possible damage induced by an UVA light used for disinfection. But, if the light is used for disinfection, it is mandatory to reflect the microorganism that is intended to disinfect, and the minimum dose needed (including some reference). This is mandatory because it would not have sense to study the effect of the UVA light if the dose provided by the light is below the minimum dose required to achieve disinfection of the intended microorganism.

So it is mandatory for the authors to reflect in the text, the dose needed to disinfect the target microorganisms, and the dose provided by the UVA during the experiments. If the provided dose is below the disinfection dose, the study of the cell damage will be useless from the practical point of view.

In lines 95-96 the authors wrote “The safety of using UV-A light in the operating room is not known, thus our study was required to begin defining the safety profile of intraoperative UV-A exposure.” I think that the authors should read and include in the references the following documents.

https://www.icnirp.org/cms/upload/publications/ICNIRPUV2004.pdf

https://www.icnirp.org/cms/upload/publications/ICNIRPUVWorkersHP.pdf

In Methods section, when describing the UV light sources it is mandatory to include the spectrum of the sources.

In line 130-131 it is not clear which part of the murine model was exposed to UVA light. Please clarify.

In line 225 there is a typographical error

In line 261-262 the authors report the industry photobiological standard limit of continuous UVA exposure to 10W/m2. Please include the source of this data, and explain what “continuous exposure” means.

6. PLOS authors have the option to publish the peer review history of their article (what does this mean?). If published, this will include your full peer review and any attached files.

Reviewer #1: No

Reviewer #2: **Yes: **Justo Arines

---

## [Author Response · Author response to Decision Letter 0]

30 Jun 2023

Reviewer #1: Authors investigated the safety of UVA-lighting system in an operative room in vitro and in vivo study of porcine model, concluding that UVA irradiation in an OR might be safe for patients and medical personnel. It is certainly promising results, but some major points need to be clarified.

Major points:

1. Authors should show the wavelength specification of UVA lamps they used in this study. The information is including wavelength distribution, peak wavelength, with filtration system or not? etc. If it contains UVB ranged wavelength that have surely harmful potential effects as genotoxicity. Furthermore, most important point is that UVA lamps actually has germicidal efficacy? Authors should show the efficacy data of lamps both UVA and UVC lamps they utilized in this study.

We added description of the types of emitters used, as well as full spectra in new Figure 5.

We added log-reductions expected for the UV-A and UV-C doses used in this study to the end of the discussion section where the cited UV-A efficacy studies were discussed.

2. DNA damage evaluation; UVA could induce a number of ROS related modification, as authors described in the text, including DNA damage as 8-oxoguanie (8-oxoG) that could potentially induce genotoxicity. It means that the evaluation of 8-oxoG staining after UVA exposure is more important than comet assay which commonly assayed direct DNA damage by UVB or UVC.

The authors thank this reviewer for their comments. To address the potential impact of UV-A on oxidative DNA damage through 8-oxoG modifications, we investigated the expression of OGG1, the primary enzyme responsible for recognizing and removing oxidized guanine base pairs. Specifically, we made the following changes to the manuscript:

1. The following was added to the introduction: “It has also been reported that UV-A can cause DNA damage by inducing oxidative modifications such as 7,8-dihydro-8-oxoguanine (8-oxoG) (8–10). Oxidixed guanine, 8-oxoG, can be repaired through the DNA base excision repair pathway (3–5). An enzyme named 8-oxoguanine DNA glycosylase 1 (OGG1) recognizes and excises 8-oxoG modifications (3–5). OGG1 protects DNA integrity and decreases in OGG1 expression have been associated with oxidative DNA damage (3,6,7).”

2. A western blot probing OGG1 expression was added to Figure 1:

Figure 1. Western blot comparing markers of DNA repair. (A) HEK293 cells exposed to 10 minutes of 10 Gy radiation as a positive control or UV-C light for 2 hours had detectable amounts of CHK1 phosphorylation (pCHK1). Phosphorylation of histone H2A.X (pH2A.X), the lower 15 kDa band, was highest in the 10 Gy positive control sample. pH2A.X was higher in cells exposed to UV-C for 2 hours compared to cells exposed to either UV-A or normal LED light (control). (B) A decrease in OGG1 expression is associated with oxidative DNA damage. Cells exposed to either UV-A or UV-C light for 1 hour had slight increases in expression of OGG1. Exposure to normal LED light (control), UV-A, and UV-C for 2 hours reduced OGG1 expression compared to 1-hour exposure. Cells exposed to 10 minutes of 10 Gy gamma radiation (positive control) had the most significant decrease in OGG1 expression.

3. The following description of the OGG1 western blot was added to the results: “A reduction in the expression of OGG1 has been reported to be correlated with an increase in oxidative DNA damage (6) (7). Since UV-A exposure has previously been shown to induce oxidative DNA damage, we investigated the expression of OGG1 in HEK-293 cells exposed to either normal LED light (control), UV-A, or UV-C light for either 1 or 2 hours (Figure 1). After 1 hour, cells exposed to UV-A or UV-C had slight increases in OGG1 expression compared to normal LED light (control). Exposure to normal LED light (control), UV-A, and UV-C for 2 hours reduced OGG1 expression in each group compared to the 1-hour exposure groups. Interestingly, cells exposed to normal LED light as a control had the greatest reduction in OGG1 expression, followed by UV-C, and then UV-A had the least reduction in OGG1. As a positive control, cells exposed to 10 minutes of 10 Gy gamma radiation had the most significant decrease in OGG1 expression.”

4. We also updated Figure 1 to include the 1 hour and 2-hour exposures on the same western blot. We have updated the results section as: “Cells exposed to 10 minutes of 10 Gy radiation served as a positive control. The phosphorylation of CHK1 (pCHK1) was nearly undetectable in HEK293 cells exposed to UV-A light for either 1 or 2 hours (Fig. 1A). The phosphorylation of histone H2A.X (pH2A.X) is detectable as a 15 kDa band on western blot, which corresponds to the lower band in the pH2A.X blot Figure 1A. Compared to the 10 Gy positive control group, cells exposed to either UV-C light or room light (LED) for 2 hours had phosphorylation of pH2A.X as seen by positive signal from the lower 15 kDa band (Fig. 1A).”

Minor points:

3. Figure 4. “Negative stain”; Is this isotype antibody as only second antibody OR no exposure of either UV lamps?

This experiment was entirely redone to address high background levels. We performed immunofluorescence instead of immunohistochemistry with DAB chromagen, per recommendation by the manufacturer of both primary antibodies (6-4PP and CPD). The results of immunofluorescence for 6-4PP and CPD on mouse livers exposed to either room light (LED), UV-A, or UV-C are included in Figure 3. Additionally, the methods section was updated as follows: “Immunofluorescence for DNA damage-related photoproducts. 8-week-old C57/BL6 mice were humanely euthanized and then underwent liver resection. Resected livers were exposed to either room light (LED), UV-A light at 30 W/m2, or UV-C light at 0.3 W/m2 for a total of 2 hours. Livers were fixed in 4% paraformaldehyde, followed by a sucrose gradient incubation and OCT embedding, prior to cryosectioning. Cryosections were permeabilized with 0.3% triton X-100 and blocked with 5% goat serum for 1 hour at room temperature. After blocking, slides were incubated with the mouse-on-mouse polymer IHC Kit (Abcam ab269452) for 1 hour at room temperature. Liver sections were then incubated overnight in a humidity chamber with primary antibody at 4oC. Primary antibodies evaluated include: cyclobutene pyrimidine dimers (1:1500, CPD, CosmoBio CAC-NM-DND-001) and pyrimidine-pyrimidone (6-4) photoproducts (1:300, 6-4 PP, CosmoBio CAC-NM-DND-002). The next day, slides were washed with PBS prior to incubating with goat anti-mouse-A488 secondary antibody (1:500, Invitrogen A11001) for 2 hours at room temperature. The slides were then washed in PBS three times before adding DAPI (1:1000) as a nuclear stain. Finally, coverslips were mounted before imaging with the 40X objective on a DM5500B microscope (Leica Microsystems).”.

4. Figure 4. 6-4PP/Room light; too high background, should be modify the immunohistochemical conditions.

The immunohistochemical experiment for 6-4PP and CPD was entirely redone to address high background levels. We performed immunofluorescence instead of immunohistochemistry with DAB chromagen, per recommendation by the manufacturer of both primary antibodies (6-4PP and CPD). The results of immunofluorescence for 6-4PP and CPD on mouse livers exposed to either room light (LED), UV-A, or UV-C are included in Figure 3. The results section has been updated as follows: “UV-A exposure does not produce DNA damage-related photoproducts in tissue. DNA damage caused by UV light exposure creates cyclobutyl pyrimidine dimers (CPD) and pyrimidine (6-4) pyrimidone photoproducts (6-4PP) (9,16,17). These photoproducts accumulate in the nucleus and can lead to cell death. We sought to determine if liver exposed to UV-A light produced CPD and 6-4PP, in comparison to either intraoperative UV-C or the normal room light (LED) condition. We used a mouse model to administer each light group to resected murine livers. Immunofluorescence was performed to detect the presence of CPD and 6-4PP in nuclei after light exposure to the liver (Figure 3). Fluorescence imaging showed UV-A light did not produce either 6-4PP (Figure 3A) or CPD (Figure 3B) in the nuclei of mouse livers, as determined by the absence of fluorescent signal. In contrast, mouse livers exposed to UV-C light did produce both DNA damage photoproducts 6-4PP (Figure 3A) and CPD (Figure 3B), confirmed by positive fluorescent signal. As a negative control, livers exposed to room light (LED) showed no positive signal for 6-4PP or CPD.”.

New Figure 3:

5. Authors should have mentioned about the latest trend of disinfection devices such as Far-UVC lamps which have a property (222nm-emission) of high germicidal efficacy and no risk of skin carcinogenesis and now be replacing by the conventional UVC disinfection system.

In the introduction, we clarified that the 254nm emission used in the study represent the most commonly used devices, cited a 222nm study as a lower-risk alternative.

Reviewer #2: The manuscript present an study on the effects of UVA and UVC radiation on DNA damage using in vitro and murine models. Analysis of the irradiance provided by an overhead UVA light developed for being used during a surgical procedure is also presented. The analysis includes data obtained at different height and data of the light reflected by a porcine model to mimic the reflection s produced during an standard surgical operation.

1. During the revision of the state of the art, I found a work (Rezaie A, et al., Ultraviolet A light effectively reduces bacteria and viruses including coronavirus. PLoS One. 2020 Jul 16;15(7):e0236199. doi: 10.1371/journal.pone.0236199. Erratum in: PLoS One. 2020 Aug 11;15(8):e0237782. PMID: 32673355; PMCID: PMC7365468.) were the authors show an study of the effects of UVA radiation on mammalian (mice) internal visceral cells. They exposure colonic mucosa to UVA and examined through endoscopy the mucosa in order to detect possible damages. Besides the authors performed tissue analysis. We think that this paper should by included in the reference section and discussion by comparing the results obtained in the present work with those included in the suggested reference.

We thank this reviewer for their suggestion. We have included Rezaie et al as a reference and included the following in our discussion: “Our results investigating DNA damage markers upon exposure to UV-A light are consistent with with a recent study that similarly explored the effects of UV-A light in human cells and murine tissue (49). In this study, Rezaie et. al found that UV-A exposure on cells did not effect viabilty or induce the DNA damage marker 8-Oxo-2'-deoxyguanosine levels, while significantly reducing a range of pathogens (49). Taken together, our data and other recent literature suggest that UV-A light does not induce a significant amount DNA damage in cells or tissues.”

2. I think that the paper needs to include data on the dose. Irradiance data is important designing the UVA light and I thank the authors for providing this data. But from a disinfection point of view, the dose is the most important data. Most of the papers related with UVA, UVB or UVC radiation provides the dose (J/m2 or mJ/cm2). So, I suggest the authors to include data on the dose used in their experiments. For calculating the dose it would by necessary to indicate the exposure time. Besides it would be interesting to know the exposure time of the tissue during a common surgical operation.

To address this comment, we have included data on dosing in our methods section. For the western blot protein expression analysis study, we included: “Cells were subsequently exposed to either 1 or 2 hours of either UV-A light at 30 W/m2 (10 times what the UV-A irradiance may be in a typical application) or UV-C light at 0.3 W/m2. This results in a dose of 10.8 or 21.6 J/cm² for UV-A and 108 or 216 mJ/cm² for UV-C.” For the comet assay and immunofluorescence studies, we included “UV-A light at 30 W/m2, or UV-C light at 0.3 W/m2 for a total of 2 hours.” For the experiment assessing reflection on surgical personnel, we included “An overhanging UV-A light fixture was set to provide irradiance of 30 W/m2 at the level of the skin” in the methods section, and “We showed that the greatest amount of reflected UV-A irradiance was 20-fold less than the patient’s exposure (0.15 mW/cm2 versus 3.0 mW/cm2)” in the results section.

3. In the same direction, I think that it is important to study the possible damage induced by a UVA light used for disinfection. But, if the light is used for disinfection, it is mandatory to reflect the microorganism that is intended to disinfect, and the minimum dose needed (including some reference). This is mandatory because it would not have sense to study the effect of the UVA light if the dose provided by the light is below the minimum dose required to achieve disinfection of the intended microorganism.

So it is mandatory for the authors to reflect in the text, the dose needed to disinfect the target microorganisms, and the dose provided by the UVA during the experiments. If the provided dose is below the disinfection dose, the study of the cell damage will be useless from the practical point of view.

We thank this reviewer for their comment. To address this concern, we have added the following to the discussion: “A safer alternative to typical UV-C light disinfection products could be devices that use low levels of UV-A light. Previous studies have shown that UV-A light is germicidal and is capable of reducing pathogens on steel surfaces (34). UV-A exposure from an overhead light source significantly reduced the amount of pathogenic microorganisms on medical equipment (34). The UV-A doses used in this study would result in approximately a 1-log10 reduction of some viruses (34,45), and up to about a 6-log10 reduction of bacteria on surfaces (33,34), and the UV-C doses used as a positive control would likewise result in a several-log reduction of many common pathogens (46).”

4. In lines 95-96 the authors wrote “The safety of using UV-A light in the operating room is not known, thus our study was required to begin defining the safety profile of intraoperative UV-A exposure.” I think that the authors should read and include in the references the following documents.

https://www.icnirp.org/cms/upload/publications/ICNIRPUV2004.pdf

https://www.icnirp.org/cms/upload/publications/ICNIRPUVWorkersHP.pdf

We have reviewed these statements and added both documents to the introduction in the sentence: “Other standards exist that may set different limits and may be commonly used in Europe or elsewhere (37,38).”

5. In Methods section, when describing the UV light sources it is mandatory to include the spectrum of the sources.

We thank the reviewer for their suggestion. We have updated the beginning of the methods section to include the spectrum of the sources: “The UV light devices were provided by GE Current, a Daintree company (East Cleveland, OH). The UV-A emitter was an array of LEDs with peak wavelength of approximately 367nm, and the UV-C emitter was a mercury lamp emitting primarily 254nm radiation. Spectral power distributions for both devices are shown in Figure 5.” Additionally, we have added a new figure (“Figure 5”).

6. In line 130-131 it is not clear which part of the murine model was exposed to UVA light. Please clarify.

We have entirely redone the experiment that was described in lines 130-131 by performed immunofluorescence for 6-4PP and CPD on mouse livers, instead of immunohistochemistry. Specifically, we have added the following information to the methods section for clarification: “8-week-old C57/BL6 mice were humanely euthanized and then underwent liver resection. Resected livers were exposed to either room light (LED), UV-A light at 30 W/m2, or UV-C light at 0.3 W/m2 for a total of 2 hours. Livers were fixed in 4% paraformaldehyde, followed by a sucrose gradient incubation and OCT embedding, prior to cryosectioning.”

7. In line 225 there is a typographical error

We have corrected this typographical error.

8. In line 261-262 the authors report the industry photobiological standard limit of continuous UVA exposure to 10W/m2. Please include the source of this data, and explain what “continuous exposure” means.

To address these comments, we have:

• Added doses for both UV-A and UV-C to the methods section. 

• Added log-reductions expected for the UV-A and UV-C doses used in this study to the end of the discussion section where the cited UV-A efficacy studies were discussed.

• Added clarification to the source of the 10W/m², as well as details on where these standards are used. References added to the ICNIRP guidelines as well as EU Directive 2006/25/EC for comparison. We added clarification that “continuous exposure” means a non-pulsed source in the context of these standards.

• Updated the description of the types of emitters used, as well as full spectra in new Figure 5.

REPSONSE TO EDITORS COMMENTS:

1. We have renamed files accordingly.

We have included the following statements: (1) “All animal experiments were conducted under institutionally approved protocols (IACUC-07733 for Sus domesticus and IACUC-21885 for Mus musculus).” (2) “8-week-old C57/BL6 mice were humanely euthanized following an institutional-approved animal protocol”, and (3) “A porcine model was used to measure the reflected irradiance of UV-A light. The animal was anesthetized per standardized protocol in a supine position.”.

"Conflicts of interest to disclose include that J. Ba Rose (JBR) received a research grant from GE Current that funded this study. Co-author Kevin Benner (KJB) is an employee of GE Current, a Daintree company, and has filed intellectual property on behalf of GE Current, a Daintree company and General Electric Company that pertains to aspects of this work."

We have added the following to the cover letter and manuscript: “Conflicts of interest: J.B.R. received a research grant from GE Current that funded this study. K.J.B. is an employee of GE Current, a Daintree company, and has filed intellectual property on behalf of GE Current, a Daintree company and General Electric Company that pertains to aspects of this work. The funders had assisted with study design, data collection and analysis, decision to publish, or preparation of the manuscript.”

We have included S1 file in cover letter and as a separately uploaded document

6. We have replaced the copyrighted images.

7. We have included captions for our Supporting Information files at the end of our manuscript.

---

## [Decision Letter · Decision Letter 1]

22 Aug 2023

Preclinical safety evaluation of continuous UV-A lighting in an operative setting

PONE-D-22-32617R1

Dear Dr. J. Bart Rose,

We’re pleased to inform you that your manuscript has been judged scientifically suitable for publication and will be formally accepted for publication once it meets all outstanding technical requirements.

Kind regards,

Sadia Ilyas, Ph.D.

Academic Editor

PLOS ONE

Additional Editor Comments (optional):

Reviewers' comments:

Reviewer's Responses to Questions

**Comments to the Author**

1. If the authors have adequately addressed your comments raised in a previous round of review and you feel that this manuscript is now acceptable for publication, you may indicate that here to bypass the “Comments to the Author” section, enter your conflict of interest statement in the “Confidential to Editor” section, and submit your "Accept" recommendation.

Reviewer #1: All comments have been addressed

Reviewer #2: All comments have been addressed

2. Is the manuscript technically sound, and do the data support the conclusions?

Reviewer #1: Yes

Reviewer #2: Yes

3. Has the statistical analysis been performed appropriately and rigorously? 

Reviewer #1: Yes

Reviewer #2: Yes

4. Have the authors made all data underlying the findings in their manuscript fully available?

Reviewer #1: Yes

Reviewer #2: Yes

5. Is the manuscript presented in an intelligible fashion and written in standard English?

Reviewer #1: Yes

Reviewer #2: Yes

6. Review Comments to the Author

Reviewer #1: Nothing further.

The authors have adequately addressed my comments raised in a previous round of review.

Reviewer #2: The authors addressed all the comments satisfactorily. They included all the information and clarifications demanded by the reviewer.

7. PLOS authors have the option to publish the peer review history of their article (what does this mean?). If published, this will include your full peer review and any attached files.

Reviewer #1: **Yes: **Makoto KUNISADA

Reviewer #2: **Yes: **Justo Arines

---

## [Editor Report · Acceptance letter]

29 Aug 2023

PONE-D-22-32617R1 

Preclinical safety evaluation of continuous UV-A lighting in an operative setting 

Dear Dr. Rose:

I'm pleased to inform you that your manuscript has been deemed suitable for publication in PLOS ONE. Congratulations! Your manuscript is now with our production department. 

Kind regards, 

on behalf of

Prof. Sadia Ilyas 

Academic Editor

PLOS ONE